# Counting Hours, Counting Losses: The Toll of Unpredictable Work Schedules on Financial Security

**Pegah Nokhiz**[*]                                                    *pegah.nokhiz@gmail.com*
*Cornell University, Cornell Tech*

**Aravinda Kanchana Ruwanpathirana**[*]                 *kanchana.ruwanpathirana@gmail.com*
*National University of Singapore*

**Aditya Bhaskara**                                              *bhaskaraaditya@gmail.com*
*University of Utah*

**Suresh Venkatasubramanian**                       *suresh_venkatasubramanian@brown.edu*
*Brown University*

**Reviewed on OpenReview:** *https://openreview.net/forum?id=PEZz2i9kiP*

## Abstract

Financial instability is a pressing concern in the United States, with drivers that include growing employment disparities and insufficient wages. While research typically focuses on financial aspects such as income inequality in precarious work environments, there is a tendency to overlook the time-related aspect of unstable work schedules. The inability to rely on a consistent work schedule not only leads to burnout and conflicts between work and family life but also results in financial shocks that directly impact workers' income and assets. Unforeseen fluctuations in earnings pose challenges in financial planning, affecting decisions regarding savings and spending, and ultimately undermining individuals' long-term financial stability and well-being.

Our objective in this study is to understand how unforeseen fluctuations in earnings exacerbate financial fragility by investigating the extent to which individuals' financial management depends on their ability to anticipate and plan for future events. To answer this question, we present a computational framework to model real-time consumption decisions under income uncertainty, drawing on advances in online planning and reinforcement learning (RL) with lookahead. We introduce a novel online algorithm that enables utility-maximizing agents to dynamically adapt consumption choices in response to financial shocks, leveraging partial deterministic information about future income. This approach forms the basis of our simulation framework, which models how workers manage consumption in the face of variable work schedules and the imperative to avoid financial ruin.

Through theoretical analysis, we quantify the utility advantage conferred by varying levels of lookahead. Empirical simulations demonstrate how increased lookahead improves financial utility. That is, with this framework, we demonstrate both theoretically and empirically how a worker's capacity to anticipate schedule changes enhances their long-term utility. Conversely, the inability to predict future events can worsen workers' financial instability. Moreover, our framework enables us to explore policy interventions aimed at mitigating the problem of schedule uncertainty. By modeling both individual behavior and potential policy interventions (e.g., advance scheduling regulations), our framework draws on ideas

---

[*]The first two authors contributed equally to this work.

from machine learning and reinforcement learning to inform economic questions surrounding information access in financial planning.

# 1 Introduction

Financial instability is a persistent challenge for many workers in the U.S., driven not only by stagnant wages and limited benefits, but also by irregular and unpredictable work schedules (Weller, 2018; Kalleberg, 2009; 2011). In sectors like food service and retail, such volatility is the norm: over 80% of workers report having little to no control over their schedules, and a majority are required to remain perpetually available for shifts (Loggins, 2020; Schneider & Harknett, 2019b). This unpredictability severely limits workers' ability to plan ahead financially, with more than half of income volatility cases traced to erratic scheduling (McCrate, 2018).

These dynamics pose a real-time computational challenge under uncertainty: individuals must make daily consumption and savings decisions in real time, while facing uncertainty about future income. Workers earn income through their employment and use it to fulfill daily necessities. How individuals allocate their income across various needs determines the utility they derive, based on how effectively they balance saving and consumption. Since income is impacted by unpredictable work schedules, the challenge becomes one of optimizing consumption policies without full information about future earnings. This motivates a technical inquiry into quantifying the utility loss from income volatility and systematically evaluating the impact of limited foresight on financial outcomes.

Therefore, this leads to a need for a computational model that captures how individuals optimize consumption and savings in response to unpredictable schedules. The model must jointly represent the utility-maximizing behavior of agents under uncertainty and varying levels of lookahead.

Subsequently, we develop a computational model of decision-making that frames consumption under uncertainty as an online planning problem. Agents aim to maximize utility while facing stochastic income and financial shocks. Unlike classical economic models that assume full knowledge of future income (Deaton et al., 1992; Deaton, 1989), our approach models agents with partial lookahead: they receive deterministic information about income within a fixed horizon and stochastic information beyond the horizon. This formulation would allow examining how varying the amount of available future information affects utility outcomes and adaptive decision-making under uncertainty.

This modeling approach is advantageous for several reasons. First, offline models generate a singular, fixed policy incapable of adapting to new financial information/shocks. Secondly, the simultaneous occurrence of financial data availability and optimization better aligns with real-world behavior. Third, the introduction of future lookahead allows for a nuanced exploration of its utility under different conditions and parameters (i.e., various levels of lookahead). Lastly, the online framework with future information provides a robust simulation environment for studying the consequences of work schedule instability and bias in future information availability. Our model is particularly relevant in light of recent regulatory efforts aimed at improving clarity in work scheduling (e.g., the Retail Workers Bill of Rights (Golden, 2015)), which aim to provide workers with more predictable incomes. By empirically exploring policy-driven interventions (e.g., increasing lookahead), we offer insights into how technical tools can evaluate and potentially inform policy decisions.

**Technical Framing and ML/RL Positioning.** Our setup connects naturally to work in reinforcement learning (RL) and machine learning (ML), where lookahead is a well-established technique for improving sequential decision-making under uncertainty by estimating the long-term consequences of actions, rather than relying solely on immediate outcomes. More formally, RL often studies $H$-step lookahead, where the agent optimizes its actions by considering the expected utility over $H$ steps (Sikchi et al., 2022). This paradigm has been extensively studied, with numerous works examining the role of lookahead with fixed horizons (Moerland et al., 2020; Sikchi et al., 2022; Dalal et al., 2021; Efroni et al., 2018). We explore a related but distinct form of lookahead. Rather than evaluating expected utility over a fixed horizon, we consider agents who are given exact knowledge of events within a fixed horizon and stochastic information beyond it, which they use to guide decision-making.

Recent work by Merlis et al. (2024); Merlis (2024) also explores a similar concept of lookahead in RL with partial deterministic lookahead, providing algorithms with theoretical guarantees, when the agents are informed of future rewards or transitions before acting. While their focus is on core RL components like rewards and transitions, our work brings this idea to a new domain: economic decision-making under uncertainty, where the lookahead instead pertains to external environmental parameters such as income.

These studies situate our setting within a broader space of lookahead models, distinguishing traditional RL (where future outcomes are estimated) from settings like ours, where future information within the lookahead horizon is known exactly. Our proposed online planning algorithm builds on this refined lookahead structure and underpins the paper's main technical results.

**Contributions.** Overall, the main contributions of this work are:

- **Online Planning Algorithm with Deterministic Lookahead:** We introduce an online planning algorithm that models an agent's sequential consumption decisions under stochastic income realizations and varying degrees of deterministic future information. The algorithm recomputes a consumption policy at each timestep based on newly revealed income and shock information. This dynamic re-planning framework accommodates real-time updates and simulates bounded lookahead in environments with financial uncertainty.

- **Theoretical Characterization of Lookahead Utility Gaps:** We formally characterize the utility gap between agents with different lookahead horizons, showing that the expected cumulative utility increases monotonically with the size of the lookahead. Under reasonable assumptions (e.g., concave utility functions and bounded shock distributions), we derive lower bounds on this gap and demonstrate tightness of the bounds.

- **Empirical Evaluation Under Realistic Constraints:** We empirically evaluate the impact of lookahead on cumulative utility through simulations that incorporate practical constraints such as income shocks and minimum subsistence (i.e., payments for basic needs like food and shelter). We explore scenarios in which agents differ in lookahead horizon and income volatility, evaluating the marginal value of lookahead information.

- **Temporal Effects of Information Asymmetry:** We investigate how asymmetric access to future information leads to measurable differences in optimal decision-making trajectories, with a focus on how these effects particularly affect those representing socioeconomically disadvantaged groups.

- **Mitigation Strategies:** We investigate mitigation strategies inspired by regulatory interventions such as fair workplace laws to examine how the adverse impacts of just-in-time scheduling on utility, can be reduced. Our experiments evaluate the effectiveness of these interventions within our framework, exploring how policy-driven constraints on scheduling influence worker well-being.

**Note on Terminology.** Before moving to the next sections, we briefly clarify a point of terminology. While we could refer to our approach as "online learning", it more closely resembles "planning", where actions are dynamically selected in response to evolving states, rather than the traditional machine learning notion of incrementally updating a model as new data arrives.

## 2 Related Work

**Work Schedule Instability.** The current focus of research primarily lies in the field of sociology, specifically examining irregular work scheduling and its various repercussions. Unstable schedules cause income volatility (Hannagan & Morduch, 2015; Morduch & Siwicki, 2017; Farrell & Greig, 2016; Reserve, 2016; Schneider & Harknett, 2017) and income volatility results in financial and life hardship (Bania & Leete, 2006; Reserve, 2016; Leete & Bania, 2010; McCarthy et al., 2018; Lambert et al., 2019). This encompasses issues such as burnout from precarious work schedules (Schneider & Harknett, 2019a; Hawkinson et al., 2023) and work-family conflicts (Golden, 2015; Henly & Lambert, 2014), particularly affecting parents with unpredictable or

just-in-time schedules. The impact also extends to areas like anxiety and child behavioral problems linked to parental work instability (Schneider & Harknett, 2019b). Additionally, the field of Human-Computer Interaction (HCI) has also strived to study similar repercussions with a participatory outlook (Uhde et al., 2020; Wood, 2021; Lee et al., 2021). They investigate the necessity of worker participation in deciding schedules to ensure fairness (Uhde et al., 2020) and how to use elicitation methods to figure out worker preference models to assist in schedule management (Lee et al., 2021).

Further, statistical data shows a pronounced unfairness in advance notices for altering work timetables for underprivileged groups. Hourly workers, individuals with lower educational attainment, women of color, and specific service sectors are disproportionately affected (Schneider & Harknett, 2019b; McCrate, 2018) with managerial discretion (Wood, 2018; Lambert, 2008). These work schedules make it difficult to plan for the future (Shah et al., 2015) and difficulty in planning would result in disproportionate financial poverty and hardship (Gennetian & Shafir, 2015). Along the same lines, there are some reports pointing to scheduling software and planning algorithms as a factor behind more unpredictable scheduling, particularly for low-wage workers in the service industry (Kantor, 2014; Lancaster, 2015; Griesbach et al., 2019; Zhang et al., 2022). For example, a New York Times article pointed out how some employees with algorithmic schedules rarely learned their timetables more than three days before the start point of a workweek (Kantor, 2014) or how pay reduction is correlated with sudden schedule changes and sales figures (Loggins, 2020).

**Simulation.** Simulation has been used in many social settings (Nokhiz, 2024), including fairness in lending (Liu et al., 2018), resource allocation (Ensign et al., 2018; Elzayn et al., 2019), college admissions (Hu et al., 2019; Kannan et al., 2019), financial analysis (Cristelli et al., 2011), technology adoption (Klügl & Kyvik Nordås, 2023), studying supply chain shortages (Yongsatianchot et al., 2023), and simulations of global crisis like the pandemic to minimize the spread of virus (Abe et al., 2022). To study the effects of advance notice on individuals, we follow in the path of research by D'Amour et al. (2020), Zheng et al. (2020), and Nokhiz et al. (2021; 2024); Nokhiz (2024), which use simulation as a mechanism to study behaviors of agents in systems.

**Consumption Models.** Consumption models fall within the broader category of discounted utility (DU) models that collectively contribute to our understanding of how individuals navigate decisions related to consumption and saving over time (Deaton et al., 1992; Deaton, 1989). These models involve individuals engaging in discounted consumption utility maximization, where they generally prefer immediate rewards over future rewards of the same size (Chabris et al., 2010). The decision-making process involves choices regarding when to consume or save (Samuelson, 1937). Several widely used intertemporal consumption models include the permanent income hypothesis (PIH), the life-cycle model (Deaton et al., 1992; Friedman, 2018; Parker, 2010), and the neoclassical consumption model (Bütler, 2001). The income fluctuation problem (IFP) is also a related consumption model with an infinite time horizon optimization with income uncertainty and an upper bound on consumption (the upper bound hinders any consumption more than the amount of assets individuals own, i.e., one cannot consume more than what they possess) (Ma et al., 2020; Sargent & Stachurski, 2014; Deaton, 1989; Den Haan, 2010; Kuhn, 2013; Rabault, 2002).

**Online Learning in Investment.** In investment, where high volatility and real-time data prevail, an online model is needed to allocate investments across assets and maximize cumulative wealth through sequential optimizations. This application of online learning, known as online portfolio selection (Dochow, 2016; Xi et al., 2023; Li & Hoi, 2014; 2018), is an algorithmic trading strategy that predicts future prices of risky assets using historical data. Online learning algorithms then optimize the portfolio with loss functions tailored to financial objectives, aiming for maximum wealth.

While investment models (Bayraktar & Young, 2012; Grandits, 2015) have valuable components like modeling of uncertainty, they can only imperfectly model consumption. They focus on portfolio allocation between risky and risk-free assets rather than consumption decisions. Although individuals can adjust allocations to avoid financial ruin, this has no direct equivalent in everyday saving and spending choices. Moreover, investment models do not impose consumption constraints.

**Reinforcement Learning and Lookahead.** The concept of lookahead is well-studied in reinforcement learning (RL), where it typically refers to planning over a fixed horizon $H$ by considering the expected outcomes of action sequences within that horizon (Sikchi et al., 2022). In this setting, the agent optimizes

policies by evaluating the expected cumulative reward over $H$ future steps, relying on a known or learned model of environment dynamics (Moerland et al., 2020; Sikchi et al., 2022; Dalal et al., 2021; Efroni et al., 2018). Crucially, this lookahead involves estimating expected outcomes based on the stochastic dynamics of the environment. In contrast, our formulation of lookahead assumes deterministic knowledge of future environmental parameters, such as income shocks, within a limited horizon, with stochastic uncertainty only beyond this horizon. Unlike the RL setting, where lookahead explores expected consequences over states and actions, our model works directly with known future parameters over the lookahead horizon.

Recent work by Merlis et al. (2024); Merlis (2024) considers partially deterministic lookahead in RL settings, where agents have access to exact future information about rewards or state transitions before acting. These works analyze performance guarantees such as competitive ratios and regret bounds. However, their focus is on one-step lookahead with explicitly defined states and actions, and their lookahead applies to rewards or transitions. While related, our work differs in that we consider deterministic lookahead over varying horizon lengths, focusing specifically on agents' consumption decisions under stochastic income. Our lookahead is on environmental parameters rather than rewards or transitions, and our goal is to understand the behavioral effects of different lookahead horizons, rather than to optimize policies in a general RL framework. Therefore, our claims, framework, and methodological approach differ significantly from those of Merlis et al. (2024); Merlis (2024).

## 3 Determining Consumption and Savings: An Adaptive Planning Algorithm

A crucial component of a framework designed to investigate the significance of advance notice (future lookahead) involves creating a model that captures consumption under lookahead while accommodating uncertainty – specifically, how individuals, facing financial uncertainty, make decisions regarding the amount to consume and save. Although various models attempt to represent consumption under uncertainty, they all fundamentally rely on the concept of discounted utility, which is the most common model in economics for understanding the interplay between consumption and savings. In a discounted utility model, the agent, at each time step, consumes a certain amount $c$ and receives utility $u(c)$ from some concave function $u$. The objective is to devise a policy to determine a consumption value $c$ in a way that maximizes the total discounted utility. After this brief introduction, we will formally articulate the specific cases we study, assumptions, and models in the following section (§3.1).

### 3.1 Our Models

In this section, we introduce the main models that we study in this work. We consider both deterministic and stochastic models that use a discount factor $\beta$ to compute the utility. We assume that time is discretized into integer steps, and let $T$ be the effective overall job timeframe, i.e., the algorithm's time horizon. Let $a_t, c_t, y_t$ be the assets, consumption, and income at time $t$, respectively. We also let $R_t$ denote the gain from assets (independent of the size of the assets), i.e., the appreciation/depreciation rate of assets. Let $u(.)$ (in our context, we employ $u(c) = \sqrt{c}$ which is in the class of isoelastic utility functions that are commonly used in macroeconomics) be the utility. Let $\mathcal{D}_Y$ be the income distribution and $\mathcal{D}_R$ be the returns distribution.

In all the models, the goal of the algorithm is to maximize the total utility. More formally, it is to solve the following optimization problem:

$$\max_{c_1,\dots,c_T} \sum_{t=1}^{T} \beta^{(t-1)} u(c_t) \tag{1}$$
$$\text{subject to: } a_{t+1} = R_t(a_t - c_t) + y_t \text{ and } 0 \le c_t \le a_t$$

The constraint $0 \le c_t \le a_t$ ensures that the worker consumes from the assets available to them. This constraint ensures the worker could consume without going to ruin. The model equation $a_{t+1} = R_t(a_t - c_t) + y_t$ shows how assets evolve over time given the consumption and income.

**The offline or deterministic model.** In the offline model, the income and return values $y_t, R_t$ are known a priori to the algorithm (as, of course, is the starting asset value, $a_1$). This is the most common model

considered in traditional economic theory, and the solution can be found using dynamic programming. The "states" in the dynamic programming correspond to the time step of interest and the total assets remaining.

**The stochastic model.** In the stochastic model, the income and return values are stochastic and they come from known distribution. In this case, an algorithm can optimize the modified objective:

$$\max_{c_1,\ldots,c_T} \sum_{t=1}^{T} \beta^{(t-1)} \, \mathbb{E}(u(c_t)) \tag{2}$$

The parameters and the constraints follow the same definitions as in the deterministic case, with the caveat being that $y_t, R_t$ are updated using the "realized" values (not their expectations). Note that this is already an online algorithm: at every $t$, the algorithm computes the consumption value using the expected $y, R$ for the future, but as the $y_t, R_t$ values get updated, the algorithm changes its behavior.

Finally, to explore the effects of future information (lookahead), we consider a combination of the two models.

### 3.2 Online Consumption Algorithm with Lookahead

We finally consider a model that has "limited determinism" controlled by the extent of lookahead, and the rest of the process is stochastic. Formally, at any time $t$, the algorithm knows the exact values of $y_{t+i}$ and $R_{t+i}$, for $i = 1, 2, \ldots, \tau$, for some lookahead parameter $\tau$.[1] Additionally, it knows the distribution of the parameters for later time steps. Once again, in contrast to offline economic models, our online model finds an adaptive consumption policy that changes over time given the lookahead as more information arrives.

The algorithm itself is a straightforward combination of the deterministic and stochastic cases. At each step, the algorithm computes:

$$\max_{c_1,\ldots,c_T} \sum_{t=0}^{\tau} \beta^t u(c_t) + \mathbb{E}\left( \sum_{t=\tau+1}^{T} \beta^t u(c_t) \right) \tag{3}$$

where $\mathbb{E}$ is the expectation over the stochasticity in income and returns beyond the lookahead horizon.

Note that the algorithm incorporates new data (as well as lookahead information) as it arrives, which is why it is an online algorithm. For simplicity, we assume that a lookahead of $\tau$ implies the agent is aware of the exact return and income values for the next $\tau$ time steps, and for the time steps beyond $\tau$, distributions of these parameters are known.

For this model, we define Algorithm 1. This algorithm solves a stochastic dynamic programming (DP) at the start (in Line 2) where it populates the table $V$ (of size $* \times T$ where $*$ is a placeholder for the size of the assets, with dynamic resizing as the bounds for the assets increase) with the maximum utility values for each asset and time point pair in the possible paths of the stochastic system. The algorithm uses backward induction over a discretized space using the known distributions of income and assets to populate the value table. With this in hand, at time point $t$, the goal is to run a deterministic DP in Line 4 using the precalculated table $V$ and use that to calculate the optimal consumption at that time point $t$. This gives us an online algorithm that yields a new consumption at each time point $t$, based on the set of historical choices as well as the lookahead information available.

## 4 Theoretical Analysis

In this section, we establish a few theoretical results on the consumption behavior and the "power of lookahead". We analyze the advantage that a worker privileged with lookahead, could attain compared to an underprivileged worker without lookahead. Let $y_t, a_t, c_t$ be the income, assets and consumption at time $t$. Let $T$ be the timeframe of the job and $u(c) = \sqrt{c}$ be the utility gained from consuming $c$.

---

[1]It is also interesting to consider further hybrid models; e.g., the algorithm has a lookahead over $y_t$, but not over $R_t$. This is also realistic in practice since return rates are governed by the market while income lookahead can be controlled by the employer. We do not consider such models in this work.

---

**Algorithm 1** Online consumption with lookahead

---

1: Let $\tau$ be the lookahead, $a_0$ be the initial assets
2: Solve the following optimization problem and save the maximum utility for each $a, t$ in a table $V$ where $V[a, t]$ gives the utility of $a, t$ pair,

$$\max_{c_1, \ldots, c_T} \mathbb{E} \left( \sum_{t=0}^{T} \beta^t u(c_t) \right)$$

$$x_{t+1} = R_t(a_t - c_t) + y_t \text{ for all } t \ (y_t \in \mathcal{D}_Y, R_t \in \mathcal{D}_R \text{ for } t \in \{0, \ldots, T\})$$

3: **for** $r = 0$ to $T$ **do**
4:     Solve the following optimization problem to get $c_r$ given $a_r$ using dynamic programming,

$$\max_{c_1, \ldots, c_T} \sum_{t=r}^{r+\tau} \beta^t u(c_t) + V(a_{\tau+1+r}, \tau + 1 + r)$$

$$a_{t+1} = R_t(a_t - c_t) + y_t \text{ for all } t \ (y_t, R_t \text{ exactly known for all } t \in \{r, r+1, \ldots, r+\tau\})$$

5:     Set $a_{r+1} = R_r(a_r - c_r) + y_r$
6: **return** The sequence of $c_t$s at each time $t$

---

Our first result shows that even in the very simple setting of $\beta = 1$ (no discount factor), and income $y_t$ in the range $[0, Y]$, the difference between the total utility of algorithms with a $k$-step lookahead and without lookahead can be $\Omega(k\sqrt{Y})$. This is true not only of the algorithms we study, but *any* algorithm. This indicates that there are instances where a worker with lookahead privileges gains an edge over an underprivileged worker without lookahead, and the advantage grows linearly with the level of lookahead.

**Theorem 1.** *Consider two individuals, one with a lookahead of $k$ steps and one with no lookahead. Let $c_1, c_2, \ldots, c_T$ be the consumption of the individual with lookahead $k$ and $z_1, z_2, \ldots, z_T$ be the consumption of the individual with no lookahead. There exist instances where all incomes are in the range $[0, Y]$, such that*

$$\sum_{t=1}^{T} \sqrt{c_i} - \sum_{t=1}^{T} \sqrt{z_i} \geq \Omega(k\sqrt{Y}) \tag{4}$$

While our lower bound result is strong for large values of $k$, it is not very strong for small values. This is partly because we want to emphasize that a gap that grows with $k$ holds even with $R_t = \beta = 1$.

We demonstrate that for any lookahead $k$, there exists an income distribution (dependent on $k$) such that the agent with lookahead achieves a $\Omega(k)$ advantage in utility over the agent without lookahead. This dependence on $k$ does not weaken the claim, as our goal is merely to establish the existence of such instances.

*Proof.* As discussed above, we consider a very simple setting: $\beta = 1$, returns $R_t = 1$ for all $t$. Further we will assume that the individuals start with $a_1 = 0$ (no initial assets).

We consider the following input. Let $Y$ be any parameter $> 0$. Let $y_t = Y$ for $t \leq k/2$ and $y_t = x \cdot Y$ for $k/2 < t \leq k$ where $x$ is a value uniformly sampled from $[0, 1]$. Note that both the individuals (the one with and without lookahead) know this input distribution. For simplicity, we also assume that the total time horizon $T$ equals $k$ (this assumption can be easily removed by setting $y_t = \frac{(1+x)}{2}Y$ for all $t > k$). As a final simplification, since the incomes can all be scaled, we can assume that $Y = 1$ for the remainder of the proof.

First, consider the individual with $k$ lookahead. Note that they can see the value of $x$, and thus they can consume an amount $\frac{(1+x)}{2}$ at every time step. This is feasible because the first $k/2$ steps have income $y_t = 1$, and so the assets remain above the consumption at all time steps. This yields a total utility of $k\sqrt{\frac{1+x}{2}}$.

Now, consider an individual who does not have any lookahead. Intuitively, they cannot guess the value of $x$, and thus consuming $\frac{1+x}{2}$ is infeasible. But note that the individual may use a complex (possibly randomized) algorithm that consumes non-uniformly and achieve a high utility. We show that this is not possible.

The starting point of the proof is the classic minmax theorem of Yao (Motwani & Raghavan, 2013): for a given input distribution, an optimal algorithm for inputs drawn from this distribution, is a deterministic algorithm. In other words, in order to prove our desired lower bound, it suffices to restrict ourselves to deterministic algorithms and prove a bound on the difference in total utility, in expectation over the choice of $x$. For any deterministic algorithm, since in time steps $1, \ldots, (k/2)$, the algorithm only sees income of 1, the values consumption $z_1, z_2, \ldots, z_{(k/2)}$ are all fixed. Let $S = z_1 + z_2 + \cdots + z_{(k/2)}$.

First, suppose it so happens that

$$\left| S - \frac{k}{2} \cdot \frac{1+x}{2} \right| > c \cdot k, \tag{5}$$

for some parameter $c$. In this case, we will argue that $\sum_i \sqrt{z_i}$ is significantly smaller than $k\sqrt{\frac{1+x}{2}}$. The starting point is the following inequality about the strict concavity of the square root function:

**Lemma 1.** *Let $a \in (1/2, 1)$ be a constant, and let $w \in (0, 1)$. Then we have:*

$$\sqrt{w} \leq \sqrt{a} + \frac{1}{2\sqrt{a}}(w - a) - \frac{1}{8}(w - a)^2. \tag{6}$$

The proof follows by a simple calculation.

*Proof.* We have:

$$\sqrt{w} - \sqrt{a} - \frac{1}{2\sqrt{a}}(w - a) = (w - a)\left( \frac{1}{\sqrt{w} + \sqrt{a}} - \frac{1}{2\sqrt{a}} \right) = \frac{(w - a)(a - w)}{2\sqrt{a}(\sqrt{w} + \sqrt{a})^2} = -\frac{(w - a)^2}{2\sqrt{a}(\sqrt{w} + \sqrt{a})^2} \tag{7}$$

The denominator is $\leq 8$, and thus by rearranging, the inequality follows. $\square$

Now, let us write $a = \frac{1+x}{2}$. By assumption, we have that $|z_1 + z_2 + \cdots + z_{k/2} - \frac{k}{2}a| > ck$, and thus we have

$$\sum_{i \leq k} |z_i - a| > ck. \tag{8}$$

Next, we can use Lemma 1 to conclude that

$$\sum_{i \leq k} \sqrt{z_i} \leq k\sqrt{a} + \frac{1}{2\sqrt{a}} \sum_{i \leq k} (z_i - a) - \frac{1}{8}(z_i - a)^2. \tag{9}$$

Now, since the consumption cannot be larger than the overall income (which is $ka$), the middle term on the RHS is $\leq 0$. Thus, we have

$$\sum_{i \leq k} \sqrt{z_i} \leq k\sqrt{a} - \frac{1}{8}(z_i - a)^2. \tag{10}$$

Next, by the Cauchy-Schwartz inequality and equation 8,

$$\sum_i (z_i - a)^2 \geq \frac{1}{k} \left( \sum_i |z_i - a| \right)^2 > c^2 k. \tag{11}$$

Together, the above inequalities imply that $\sum_i \sqrt{z_i} \leq k\sqrt{a} - \frac{c^2 k}{8}$. This shows that if the values $z_i$ chosen by the deterministic algorithm satisfy equation 5, then the algorithm with no lookahead has total utility $\Omega(k)$ worse than an algorithm with lookahead.

The final step is to prove that if $x$ is chosen at random from $(0, 1)$, the condition equation 5 holds with a constant probability for some $c > 0$. Since $S$ is fixed, the condition is equivalent to $|\frac{2S}{k} - \frac{1+x}{2}| > 2c$. Equivalently, $|\frac{4S}{k} - 1 - x| > 4c$. For any fixed $\alpha$, if $x \sim_{\text{uar}} (0, 1)$ the probability that $|\alpha - x| \leq 1/3$ is clearly $\leq 2/3$. Thus, the condition above must hold with $c = 1/12$, with probability at least $1/3$.

Therefore, we have that with probability $1/3$, the no-lookahead algorithm is $\Omega(k)$ worse than the algorithm with lookahead, and it can never be better. Thus the *expected* difference between the total utilities is also $\Omega(k)$. Yao's minmax theorem implies that this bound holds for any (possibly randomized) algorithm. $\square$

Theorem 1 shows that an algorithm with lookahead has a provable advantage over an algorithm that knows only the distribution of the incomes, even in the simplest setting where the decay factor $\beta = 1$ and the returns are all 1. Furthermore, the advantage grows *linearly* with the amount of lookahead. In particular, if an individual has infinite lookahead, they can have an advantage of $\Omega(T)$.

Next, we show that when $\beta = 1$ and $R_t = 1$ for all $t$, the lower bound from Theorem 1 cannot be improved.

**Lemma 2.** *Suppose $\beta = 1$ and the return $R_t = 1$ for all $t$. Let $y_1, y_2, \ldots, y_T$ be a sequence of income values with $y_t \geq 0$ for all $t$. Then if the consumption sequence of a $k$-lookahead algorithm is $\{c_1, c_2, \ldots, c_T\}$, there exists a no-lookahead algorithm whose total utility is $\sqrt{c_1} + \sqrt{c_2} + \cdots + \sqrt{c_{T-k}}$.*

In other words, the difference in the total utility (between the $k$-lookahead and the no-lookahead algorithms) is simply $\sqrt{c_{T-k+1}} + \cdots + \sqrt{c_T}$. In settings where all the $c_t$ are of magnitude $O(\text{income})$, this corresponds to $O(k)$ times the square root of the income, which is the lower bound in Theorem 1.

*Proof.* The proof is simple: a no-lookahead algorithm can mimic a $k$-lookahead algorithm, but with a delay of $k$ steps. We will call this the $k$-delay algorithm. Let $c_1, c_2, \ldots, c_T$ be the consumption squence of the given $k$-lookahead algorithm. The $k$-delay algorithm is defined as follows,

> For $t = 1$ to $T$:
> 1. If $t \leq k$, consume 0
> 2. Else consume $c_{t-k}$

The total utility bound is easy to see. One only needs to check that the algorithm is feasible (i.e., it satisfies the condition that the total consumption until time $t$ is bounded by the total income plus the assets until that time). This is easy to check because the algorithm consumes 0 for the first $k$ steps, while the income $y_t \geq 0$. Since the decay factor $\beta = 1$, delay does not change the utility the algorithm receives. $\square$

**Remark.** We see that the proof relies on having $\beta = 1$. If we take into account factors such as inflation (e.g., $\beta = 0.95$), then the "power of lookahead" can likely be made much more significant.[2] The agents with lookahead could potentially use the lookahead information to anticipate future gains and consume more than the expected consumption and therefore gain a significant advantage over the agents without lookahead.

## 5 Experiments

While one might argue that all the proof setup (e.g., the lookahead case considered) is not fully realistic, it serves to build intuition, while the simulations are the parts that operate in a more realistic environment.

That is, although our analysis relies on a set of simplifying assumptions and does not use fully realistic parameters, the primary objective was to gain insights into the relationship between lookahead and utility. These insights are valuable for interpreting the empirical findings, which are based on more realistic settings. The specific choices of distributions and parameters made in the theory were to ensure the problem remained analytically tractable.

Therefore, in this section, using our algorithm, we explore three different sets of experiments regarding the effects of lookahead, the parameters involved in lookahead under uncertain job timetables, and mitigation schemes.

### 5.1 Experiment Setup

We will first introduce the different elements in our experiment setup.[3]

**Workers.** Within our framework, agents represent employed individuals who earn weekly income, own assets, and decide whether to consume or save. We create an income distribution of 10,000 individuals using 2019 income data from the US Census Bureau's Annual ASEC survey of the Consumer Price Index (by

---

[2]This is a conjecture and supporting this claim is left to future work.

[3]The code can be accessed at `https://github.com/kanchanarp/CountingHours_CountingLosses_Supplementary.git`

the IPUMS Consumer Price Survey) (Flood et al., 2020; PK, 2019-2020). In the first stage, we eliminate outliers from the income distribution to address challenges related to individuals with exceptionally high or low earnings, which can be challenging to compare due to significant scale differences. We identify outliers by employing the commonly-used inter-quartile range (IQR) proximity rule (Dekking et al., 2005). Subsequently, we categorize individuals into four distinct income groups: low incomes ranging from $1.22 to $1125.49, low-middle incomes from $1125.49 to $2249.75, high-middle incomes from $2249.75 to $3374.02, and high incomes from $3374.02 to $4498.29. This classification is achieved by partitioning the overall income range into these four segments. To establish asset values, individuals are assigned the median population asset value of $123,840, derived from median percentile net worth data and median net worth by income percentile data from the Federal Reserve (of Governors of the Federal Reserve System, 2019).

*Regarding IQR:* To create a representative and analytically sound income distribution, we apply IQR method to remove extreme low and high income outliers. Income data is typically right-skewed (Benhabib & Bisin, 2018), with a small fraction of individuals earning disproportionately high incomes. The IQR method is well-suited for this type of data, as it is non-parametric and does not rely on the assumption of normal distribution (Insights, 2025; Hubert & Van der Veeken, 2008; Whaley III, 2005). Outliers are defined as values falling below $(Q1 - 1.5 \times \text{IQR})$ or above $(Q3 + 1.5 \times \text{IQR})$, where $\text{IQR} = Q3 - Q1$, effectively trimming the extreme ends while maintaining the core structure of the data. Please refer to Appendix §A.2 for additional real-world explanations, citations, and reasonings behind IQR.

**Minimum Subsistence.** Furthermore, our simulation considers minimum subsistence, i.e., the constraint that the individuals must allocate funds for their minimum basic needs, such as food and shelter (Zhang & Liu, 2020; Zimmerman & Carter, 2003; Álvarez Peláez & Díaz, 2005; Shin & Lim, 2011; Shim & Shin, 2014; Antony & Klarl, 2019; Dwork & Ilvento, 2018; Miranda-Pinto et al., 2023; 2020). This consideration acknowledges the fundamental requirement for individuals to fulfill their basic necessities as part of the decision-making process regarding consumption and utility maximization. The minimum subsistence values are derived from mean annual expenditures in 2019 from the Consumer Expenditure Surveys of the US Bureau Of Labor Statistics (of Labor Statistics, 2019), stratified based on income levels. In essence, individuals with similar income levels are obligated to cover equivalent amounts for their basic needs.

**Shocks.** Shocks, defined as alterations to an agent's financial state due to schedule instability, play a pivotal role in influencing decisions related to consumption and savings. Shocks can either positively or negatively impact income, such as sudden work-hour reductions or increases. Here, the magnitude of income shocks ranges from $-0.4$ to $0.4$, with shocks occurring as $(1 + r) \times$ income, where $r$ represents the shock value and the income represents the current value of income uniformly sampled from the income distribution. The shocks are generated from a Bernoulli process, with the shock size parameter $r$ uniformly sampled from $[-0.4, 0.4]$. We set the probability of shocks to 0.4.[4]

According to the JPMorgan Chase Institute, income volatility remained relatively consistent from 2013 to 2018, with median households experiencing an average month-to-month income fluctuation of 36% (JPMorgan Chase Institute, 2016). However, the report highlights significant variation in volatility across households and occupations, noting that income instability is both highly heterogeneous across families and inconsistent over time for the same household. The median standard deviation of income volatility is reported as 0.37. This variability is further reflected in specific job sectors. Retail workers, for instance, often encounter erratic scheduling, leading to income fluctuations of up to 50% in certain months (CLASP, 2022) (and the corresponding earnings). Gig workers, such as rideshare and delivery drivers, also face earnings volatility due to inconsistent hours and changing platform demand, with monthly income shifts typically ranging from 3.4% to 14%, depending on market conditions and algorithmic factors (Insider, 2024). Considering these patterns, we construct a realistic/reasonable shock profile as mentioned earlier for the experiments in this section.

---

[4]As an additional note, income is not a constant. It is sampled uniformly from a given distribution. Agents additionally get shocks on top of income, where the shock probability is set to 0.4.

Please refer to Appendix §A.2 for further real-world explanations, citations, and reasonings behind the adopted shock profile. Also, an extra set of experiments based on the exact shock profiles in (JPMorgan Chase Institute, 2016) are presented in Appendix B.5.

**Isolating the Impact of Lookahead and Other Parameters.** At each time point (here, a workweek), each worker within an income group shares identical observable features, like income, shocks stemming from unpredictable schedules, and returns on assets. The sole point of divergence among individuals within the same group is the extent of lookahead they possess. This deliberate design choice aims to isolate the temporal impact of lookahead on utility, eliminating other financial factors' interference. The overall job timeline spans 26 weeks, i.e., a 6-month job duration. The returns are sampled from uniformly from $[0.9, 1.1]$ and with probability 0.1 there is an additive shock uniformly sampled from $[-0.05, 0.05]$ on the return rates. The return rate ranges used here are grounded in empirical observations of *typical* historical asset performance of 5 years preceding 2020 across a broad spectrum (Carlson, 2021; Bank, 2020).

The discounting factor $\beta$ is set to the commonly-used value of 0.95. In economics, it is common to adopt a standard, reasonable value for the discount factor, like 0.95 (Patnaik et al., 2022; Cooper & Willis, 2014).[5]

Please refer to Appendix §A.2 for additional real-world explanations, citations, and reasonings behind the discount factor and return rates. Additional experiments with more granular real-world return rates and a set of different discount factors tailored to specific real-world use cases like unemployment and education can be found in Appendix B.

**Clarification on Simulation Parameters.** For completeness, we reiterate that detailed explanations of all simulation parameters, along with their real-world empirical grounding, are provided in Appendix §A.2.

### 5.2 Future Lookahead: An Empirical Inquiry

This section aims to examine the utility acquired by individuals under varying levels of lookahead. Specifically, the experiment seeks to compare those with minimal (or no) information about the future, such as protected groups like hourly, part-time workers, and specific racial groups that typically receive limited or no advance notice, against other workers with more foresight. The experiment setup is as explained in §5.1 and the outcomes of this experiment are depicted in Figure 1.

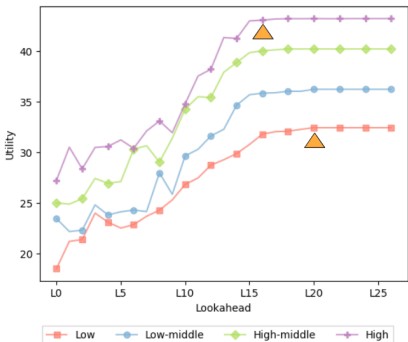

Figure 1: The final utility gained for different levels of lookahead is illustrated for four distinct income classes, each comprising 27 individuals. Workers are of similar features, with variations solely based on the temporal aspect, i.e., the amount of lookahead in their work schedules. The $x$-axis depicts the lookahead value and the $y$-axis represents the total utility at the end of $T$ steps. The two orange triangles are in place to highlight the difference in the level of lookahead needed for approaching the near-maximum utility (near-maximum utility is a utility close to that of a worker with full information at L26) as the income levels change.

**Analysis.** The insights from Figure 1 can be summarized in four key points. 1) Firstly, workers with lower lookahead and minimal information about future instability experience significantly lower financial utility

[5]https://behaviouraleconomics.jasoncollins.blog/intertemporal-choice/present-bias-examples

compared to those with higher lookahead. Unsurprisingly, individuals struggle to manage their finances effectively when confronted with unforeseen schedules. 2) Secondly, lookahead has a positive impact overall. Workers with more lookahead can efficiently manage their finances, resulting in higher utility. 3) Thirdly, workers do not require full information about their work schedules in advance. Even beyond the midpoint lookahead (lookahead 13), workers can achieve a utility comparable to those with complete information about their schedules. 4) Lastly, higher income leads to greater utility, as individuals can consume without concerns. This is evident from the consistent shift of the plots along the $y$-axis.

However, irrespective of income level, having more than the midpoint level of lookahead proves advantageous for individuals. Providing people with advance notice of their schedules can be reasonably implemented for the next 2-3 months of work without requiring employers to furnish complete information at the start of their tenure. Notably, for higher-income workers, less future information is needed to approach the utility values of someone with complete information, as observed by the leftward movement of the orange triangles across all lookahead levels.

**Robustness.** Additionally, to evaluate the model's robustness and account for variability, we run the simulation using 20 different randomly generated seeds, while keeping the original parameter settings from §5.1 consistent across all runs. Figure 2 illustrates the outcomes by showing the average values along with 95% confidence intervals as error bars, providing a measure of the reliability and consistency of the observed trends.

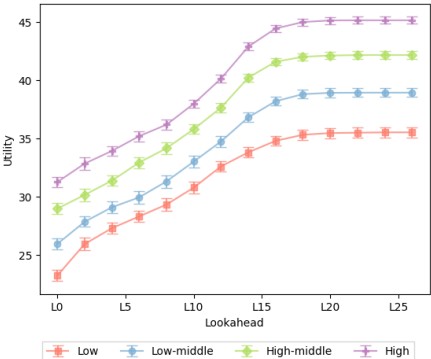

Figure 2: The main simulation in 5.2 is carried out with the original parameters in 5.1. Results are for running the simulation for 20 different randomly generated seeds. The plot displays the mean value, and the error bars display a 95% confidence interval.

**Analysis.** The key observations align with the bullet points discussed for the initial figure. That is, workers with limited foresight face significantly lower financial utility compared to those with greater lookahead, highlighting the positive impact of anticipating future schedules. While complete schedule information is not essential, having moderate foresight (e.g., lookahead of 17) yields utility close to that of full information. Additionally, higher income consistently leads to greater utility, as it allows for more flexible consumption, evident in the upward shift of utility levels across the plots. A more detailed breakdown of these results can be found in Appendix A.1.

## 5.3 Dynamics of Asset Appreciation and Depreciation

Our simulated space provides a comprehensive platform for delving into the intricacies of parameters within work scheduling. This exploration includes understanding the behavior of individuals with diminishing assets compared to those experiencing favorable returns on their savings.

Thus, this section's objective is to investigate the impact of asset appreciation and depreciation, i.e., positive and negative return rates on workers' assets on decision-making across various levels of lookahead. This also serves as an examination of scenarios wherein the workers are already at a (dis)advantage in terms of assets.

Assets depreciate when their value declines over time, influenced by factors like risky investments, fluctuating market conditions, tax obligations, possessions becoming obsolete, and wear and tear, as seen in vehicles, buildings, and cash saved without earning interest. Conversely, asset appreciation occurs when individuals receive returns on their savings (Reader et al., 2022) or make profitable investments in stocks, among other factors.

All experimental settings are similar to that of §5.1 other than the return rates on assets. Here, we compare negative return rates in the range [0.75,0.95] to positive return rates in [1.05,1.25] (Carlson, 2021; Bank, 2020).

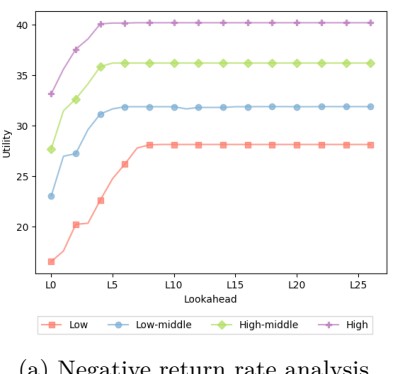

(a) Negative return rate analysis

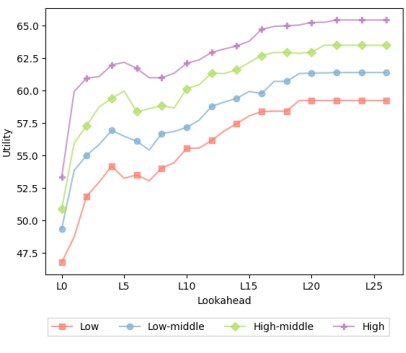

(b) Positive return rate analysis

Figure 3: Individuals with similar features but varying levels of lookahead under negative return rates ranging from 0.75 to 0.95 on their assets, as well as positive return rates between 1.05 and 1.25 on their assets.

**Analysis.** The findings are illustrated in Figure 3. Several insights emerge from this analysis: 1) Firstly, workers generally experience higher utility when they encounter favorable return rates. This is evident by comparing the utility range in the negative rates plot, which spans from 15 to 40, to the positive plot, which ranges from 45 to 65. 2) Secondly, with depreciating assets, individuals benefit significantly from small amounts of lookahead, reaching near-maximum utility. People across income classes achieve a near-maximum utility before Lookahead 10. There is a consistent decline in the lookahead value required to attain near-maximum utility, reaching around lookahead 5 for the highest income class and around 10 for the lowest income class. This observation aligns with the trend observed in the previous section, where more income classes require less future information to reach peak utility values. 3) Thirdly, in the case of positive returns, individuals have more flexibility in consumption, as they anticipate overall favorable returns ahead. Future information is not as crucial as in the negative returns scenario, where they are not at a disadvantageous situation with depreciating assets. This explains the late convergence of all income classes to the peak utility value (under positive return rates).

An additional set of experiments with more granular real-world return values is presented in Appendix §B.4.

### 5.4 Interventions

In this section, the goal is to examine the mitigating effects of two intervention scenarios. In terms of intervention policies, simulation provides a valuable sandbox environment for testing that might be challenging or even impossible to explore in the real world. In a simulated work scheduling setting, researchers, policymakers, and practitioners can experiment with various interventions, assess their effects, and fine-tune strategies without real-world consequences.

All experimental settings are similar to §5.1. We assign interventions (to a sample of 50 individuals per intervention), as follows: **(1) Compensation:** Workers will be compensated for sudden schedule changes and *on-call* shifts. This policy is modeled after the measures approved by the San Francisco Board of Supervisors, which introduced new protections for retail workers in the city, necessitating employers to provide compensation for unpredictable schedules based on factors such as employment type, hourly rate,

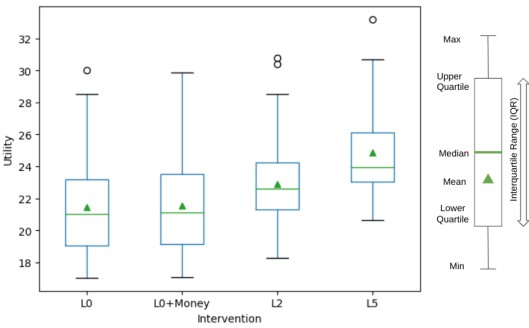 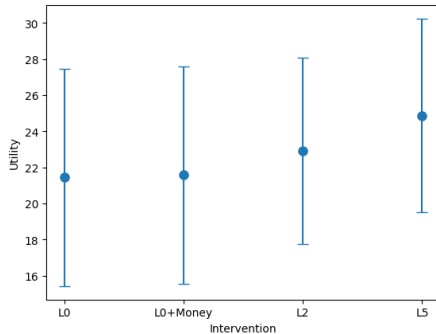

(a) Box plot of interventions          (b) Error bar plot of interventions

Figure 4: The box plot and the error bar plot of the statistical distribution of additional gained utilities for various interventions. The interventions considered are L0 + Money, which involves compensation fees for individuals with no future information; L2, assigning at least two weeks of lookahead to all; and L5, assigning at least five weeks of lookahead to everyone. In the box plot, the green triangle represents the mean, and the green line represents the median. In the error bar plot, the blue circle represents the mean and the error bars are 2 standard deviations from the mean (95% confidence interval).

hours of work (Golden, 2015). Inspired by this policy, we disburse twice the amount of earnings back when there is a negative shock. **(2) Mandatory minimum advance notice:** Every worker should be entitled to a mandated minimum lookahead, i.e., they should be aware of their schedule for the upcoming two weeks. This proposition is inspired by the Schedules that Work Act of 2014 (H.R. 5159) presented in Congress, which stipulates that if there are alterations to the schedule and minimum hours, the employer must inform the employee at least two weeks prior to the start point of the new schedule (Golden, 2015). Inspired by this policy, we assign workers two weeks and roughly one month (5 weeks) of minimum lookahead, separately.

Note that here the compensation setting is about adding a fixed constant to income when there is a shock. In these experiments, we did not explore multiple values, as the goal is to study the effects of giving a realistic fixed amount of money.

**Analysis.** Results for intervention scenarios are depicted in Figure 4 as a box plot to show the distributions (and a separate corresponding error bar plot to capture uncertainty). Three insights can be derived: 1) Firstly, interventions have a positive impact overall, evident from the increased utility across all statistical metrics (mean, median, confidence interval, and all quartiles) post-intervention. 2) Secondly, while compensation in the form of additional income for unanticipated schedule changes is beneficial, it does not substitute for providing workers with advance notice of their shift schedules. Having knowledge of future plans with the same income but a predictable schedule appears to be more effective for workers' financial well-being than an unforeseen, volatile schedule compensated with fees. A comparison between *L0 + Money* and *L2* indicates that even incorporating two weeks of lookahead is more advantageous than providing compensation for instability without any lookahead. 3) Thirdly, offering just one month of advance notice results in a notable increase in utility, as seen in the comparison between *L5* and *L0*. Even small amounts of notice can significantly enhance utility.

**Note on the Robustness of Experimental Parameters/Results.** We discussed the results for 20 different randomly generated seeds (with a 95% confidence interval) in Section §5.2, and have provided a more detailed discussion in Appendix §A.1. It is important to mention that we conducted these experiments with various random seeds, considering different runs (e.g., with a median asset of approximately $70,000 (Aurelia Glass, 2023), representing the median working-class wealth, as opposed to the population median), and with other realistic discount and return parameters, as well as varied plausible shock sizes. The overall trends in our results in the previous sections remain consistent as long as the chosen parameters fall within more realistic ranges. If one opts for more extreme and unrealistic parameters, the distinctions observed in §5.2, 5.3, and 5.4 may become even more pronounced.

**Note on Additional Real-world Scenarios.** As previously noted in several instances, to explore more diverse settings, we incorporate four additional case studies: the role of schedule instability during unemployment in §B.2, the influence of educational attainment on consumption behavior and patience for future information in §B.3, a more granular analysis of return rates by comparing cash and stocks in §B.4, and more diverse real-world shock profiles in §B.5. Across these diverse scenarios, we observe behavioral patterns consistent with those identified in the core sections. Further details can be found in the corresponding appendix sections.

## 6 Discussions and Limitations

The primary contribution of our paper lies in the development of an online framework that delves into the financial insecurity stemming from work schedule instability. This framework addresses two crucial questions pertinent to work scheduling: 1) How can we develop an online planning approach for consumption under uncertainty to formally and empirically assess the negative effects of work schedule instability on income/employment? 2) How can this in turn help with informing the development of effective mitigation policies and regulations in the workplace? We provide analytical insights into how lookahead significantly enhances individuals' utility, with a focus on the effectiveness of increased lookahead. Our empirical investigation employs simulations and explores two distinct intervention strategies aimed at mitigating the adverse effects of schedule instability. While our model is being deployed in a work setting, it applies to any scenario that involves temporal uncertainty with the possibility of improvement with lookahead. Our paper, however, has a number of limitations:

**Simulation.** A simulation is only as good as the models used to build it and relies on idealized and formal models of human behavior that are simplifications. A simplified model however is useful for an inquiry into *limits*: in our work, we show that even under ideal models of human utility-maximizing behavior, the lack of predictability and lookahead has concrete consequences for financial stability.

**Homogeneity.** The simulation assumes societal uniformity within income groups, considering temporal differences as the main variation. However, this overlooks other social inequities and heterogeneities that could impact decision-making (Nokhiz et al., 2025). Income instability can be influenced by factors such as gender (Blau & Kahn, 2017), ability (Whittaker et al., 2019), race (Zwerling & Silver, 1992), and health status (Hicken et al., 2014). Additionally, access to public policies addressing these disparities may also differ across demographic groups.

**On Algorithmic Scheduling.** In the digital age, algorithmic processes and digital technologies play a crucial role across various workforce sectors, from business process management to automated work scheduling. These tools enhance efficiency and productivity, offering advantages over manual scheduling methods by saving time and effort while providing greater control to managers and employees. However, despite these benefits, algorithmic scheduling can also introduce new challenges (Lancaster, 2015).

Scheduling analytics have the potential to enable operators to exploit their workforce unfairly. Reports from The New York Times point to scheduling software as a factor behind more unpredictable scheduling practices, particularly for low-wage workers in the service industry (Lancaster, 2015). From the standpoint of employees, this unpredictability in scheduling is perceived as a negative attribute. It is an instability enforced by employers, often utilizing workforce management technology and algorithms. Therefore, it would be beneficial to enhance our modeling approach by incorporating more provisions to account for the intricate details and mechanisms of algorithmic scheduling parameters and operations.

**Challenges in Data-Driven and Model-learning Approaches.** While our simulation framework draws on simplified models, this abstraction is intentional and grounded in well-established models in economics. These models allow us to examine the utility implications of lookahead and schedule instability in a controlled setting, using parameters informed by real-world data. Nonetheless, we acknowledge that real-world financial behavior is shaped by additional layers of complexity (like social, behavioral, psychological, occupational, and so on). A promising future direction involves integrating data-driven components such as using real-world behavioral data to learn parameters and the simulators or learning transition dynamics from observational datasets. However, such approaches face significant obstacles due to the lack of large-scale longitudinal

datasets capturing individual consumption and income patterns at sufficient granularity. Furthermore, in social systems, the dynamics are not easily observable, complicating the application of typical model-learning approaches. As such, while learning simulators from data holds potential, our current simulation approach remains a feasible and interpretable tool for exploring the impact of work scheduling instability on financial planning behavior.

**More Future Directions.** Enhancing this framework could involve integrating additional financial factors linked to temporal uncertainty, such as retirement savings constraints, differentiation between risky and riskless assets, and broader socioeconomic shifts like workplace restructuring or job relocations. Additionally, the framework offers opportunities to explore fairness and inequity in automated shift scheduling, particularly for sub-populations with sensitive attributes. It also enables research into targeted interventions within specific work environments (e.g., food and retail sectors) to mitigate the long-term effects of biased decision-making. Lastly, another additional direction for future work is to employ empirical real-world data to learn model parameters and simulate system dynamics based on observed behavioral patterns.

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

# A    More Experimental Details

This appendix section presents the robustness findings along with additional details on the simulation parameters used.

## A.1    Robustness

As mentioned in §5.2, we have included the robustness plot here as well to provide a more detailed analysis. That is, to check for robustness, we run the same original experiment mentioned in §5.2, as follows:

The experiment is conducted using the original set of parameters in §5.1, ensuring consistency across all trials. To account for variability and assess robustness, the model is executed for 20 different randomly generated seeds. The resulting Figure 5 provides a visualization of the data by displaying the mean value. The error bars are included to depict the 95% confidence interval, offering insight into the robustness of the patterns shown before.

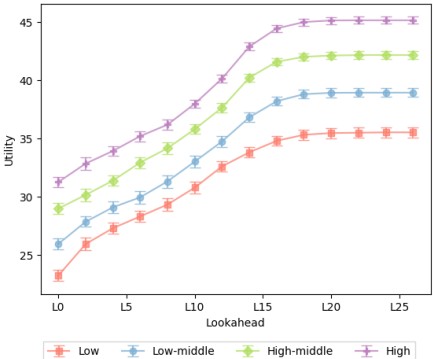

Figure 5: The simulation in 5.2 is carried out with the original parameters in 5.1. Results are for running the simulation for 20 different randomly generated seeds. The plot displays the mean value, and the error bars display a 95% confidence interval.

**Analysis.**    The insights from Figure 5 are similar to the ones in §5.2:

- The results and consumption patterns observed before are observed in this plot, which indicates robustness.

- Workers with limited foresight and minimal knowledge of future schedule instability experience notably lower financial utility than those with greater lookahead.

- Overall, lookahead has a positive effect: workers with more foresight can better navigate financial decisions, leading to higher utility.

- Full knowledge of work schedules in advance is not necessary. Even beyond the midpoint threshold (lookahead 17), workers achieve a utility level comparable to those with complete schedule information.

- Higher income directly translates to greater utility, as individuals can consume without financial constraints. This is clearly reflected in the consistent upward shift of the plots along the $y$-axis.

## A.2    More Clarifications on Simulation Parameters Used in §5.1

**Why IQR and IQR Setting.**    To ensure a representative and analytically robust income distribution, we remove extreme low and high income outliers using the Interquartile Range (IQR) method. Income data is characteristically right-skewed (Benhabib & Bisin, 2018), with a small number of individuals earning

disproportionately high incomes. The IQR method is particularly well-suited for such data because it is non-parametric and does not assume a normal distribution, unlike $z$-score-based methods, which can be distorted by skewness (Insights, 2025; Hubert & Van der Veeken, 2008; Whaley III, 2005). Specifically, per IQR's definition, the outliers are values below $(Q1 - 1.5 \times \text{IQR})$ or above $(Q3 + 1.5 \times \text{IQR})$ where $\text{IQR} = Q3 - Q1$, effectively trimming only the extreme tails while preserving the integrity of the central distribution.

**Shocks.** The parameters chosen for modeling income shocks are grounded in empirical findings from real-world data. Since the shocks are very different household by household and occupation by occupation, the common practice in economics is to adopt a reasonable range.

According to the JPMorgan Chase Institute, income volatility remained fairly stable between 2013 and 2018, with households at the median experiencing a 36% change in income from month to month on average (JPMorgan Chase Institute, 2016) (an additional set of experiments based on this shock profile is in §B.5). However, volatility levels vary substantially by household and occupation. The same report notes that income variability is highly heterogeneous, both across different families and over time for the same family. The median standard deviation of volatility is reported as 0.37.

Further illustrating this variability, retail workers often face unstable schedules that result in up to a 50% variation in work hours (and the corresponding earnings) during certain months (CLASP, 2022). Similarly, gig workers such as rideshare and delivery drivers experience changes in earnings tied to fluctuating hours and platform demand, with income shifts ranging from 3.4% to roughly 14% month-over-month depending on market conditions and app algorithms (Insider, 2024).

Taking these observations into account and to have a reasonable shock profile, we opt for income shocks that ranges from $-0.4$ to $0.4$, with shocks occurring as $(1 + r) \times$ income, where $r$ represents the shock value. The shocks are generated from a Bernoulli process, with the shock size parameter $r$ uniformly sampled from $[-0.4, 0.4]$.

**Discounting Factor $\beta$.** Similarly, the common practice in economics is to adopt a standard, reasonable value for the discount factor, like 0.95 (Patnaik et al., 2022; Cooper & Willis, 2014).[6] However, a body of research seeks to go beyond this uniform assumption by estimating more context-sensitive $\beta$ values that vary across demographic groups. For example, as discussed in the following Sections B.1, B.2, and B.3, $\beta$ can change with education level or employment status. We provide a detailed analysis of these more granular real-world scenarios in Appendix B.

**Return Rates.** The return rate ranges used in §5.1 are grounded in empirical observations of *typical* historical asset performance of 5 years preceding 2020 across a broad spectrum (Carlson, 2021; Bank, 2020). As with most economic parameters, return rates can vary significantly depending on individual circumstances and the type of asset involved. We consider typical asset returns, not highly-volatile assets. For instance, if an individual owns cryptocurrency and other very high-risk assets, they can have return rates like +215.07% or +539.96% during certain quarters and then experience very drastic drops in asset values.[7]

Therefore, like other parameters, the common practice is to choose a range that reflects realistic variability. However, as further examined in §B.4, a more granular analysis is possible by focusing on specific typical asset classes (such as stocks or cash), which allows for a more precise assessment of return dynamics. Still, these specific return rates generally remain within the broader range defined in the initial simulation.

The description of a range and the added variance of returns implies return rates uniformly distributed in a given range. With probability 0.1, we get an additive shock to the return rates where the additive amount is sampled from a different uniform distribution (Thus, *'a range and the added variance'* is not about uniform + Normal additive returns).

---

[6]https://behaviouraleconomics.jasoncollins.blog/intertemporal-choice/present-bias-examples
[7]https://www.coinglass.com/today

**Discretization of the Dynamic Programming.**   In the simulations, the asset space is discretized into integral buckets with the gap between the buckets set to 1 where the number of buckets depends on the difference between highest attainable assets and the lowest possible assets (which could be non-zero depending on the income distribution).

**Details of the Simulation Runs.**   The main results presented in §5.2 are based on a single set of simulation runs for representative individuals within each income group and for 50 runs in the mitigation experiments. This choice was intentional, as our goal was to avoid averaging across agents, which can obscure meaningful behavioral variation, especially in a setting where individual-level dynamics are of interest. As discussed in the main text, the qualitative trends in consumption and utility remain consistent across different random seeds and repeated runs. However, to meaningfully support this, we include robustness additional results across 20 random seeds in Figures 5 and 2 (similar results, but further expanded in the respective appendix section), which confirm the stability of the main patterns.

In this context, the empirical findings serve as a complement to the theoretical results established earlier, helping to illustrate broader behavioral patterns rather than focusing on fine-grained empirical fluctuations. Given the heterogeneity of agents in real-world socioeconomic environments, as shown in the detailed parameter justifications above, our interest lies in capturing these general dynamics.

## B    More Real-world Problems

In this appendix, we explore additional scenarios informed by real-world data and observed phenomena.

### B.1    Patience Factor based on $\beta$

In models of intertemporal choice, the parameter $\beta$ denotes the *subjective discount factor*, capturing the extent to which individuals prefer present consumption over future consumption. A $\beta$ value between 0 and 1 indicates that future consumption is valued less than immediate consumption, with lower values reflecting a stronger preference for the present. This formulation is closely aligned with the concept of *present bias*, which describes the tendency to overvalue immediate rewards relative to delayed ones.

A lower $\beta$ implies greater impatience, signifying that individuals are more inclined to choose smaller, immediate rewards over larger, delayed alternatives. In contrast, higher values of $\beta$ are associated with patience and a stronger orientation toward future utility.

Empirical studies on real-world events have documented that there are economic agents who exhibit more *present-focused preferences*, meaning they are more prone to choose actions that yield immediate utility when those actions affect the present, compared to when all consequences are shifted into the future (Ericson & Laibson, 2019; Lockwood, 2020). In essence, these individuals make more impulsive decisions in the short term than they would if facing the same set of outcomes at a future point in time.

The value of $\beta$ plays a critical role in determining financial behaviors such as saving and borrowing. More impatient individuals (with lower $\beta$) are typically more susceptible to accumulating debt, while those with higher $\beta$ are more likely to delay gratification and engage in future-oriented behaviors like saving. In the following examples based on real-world problems, we explore how these dynamics unfold in different contexts.

### B.2    Unemployment

This section examines one of the most prevalent real-world problems: job insecurity and unemployment. We focus on how unemployed individuals experience varying levels of utility based on their capacity for *lookahead*, that is, their ability to anticipate future events. Specifically, the analysis compares unemployed agents who have minimal or no information about the future against those with greater foresight (and the related $\beta$).

Although this paper primarily addresses dynamic work scheduling, the underlying framework also applies directly to unemployed individuals, who typically face profound uncertainty. An unemployed person may not

know: When or whether they will find a job, whether any job obtained will be full-time or part-time, how long unemployment insurance (UI) benefits will last, whether they will experience unanticipated financial shocks during the job search, how partial employment affects UI eligibility and payments, and how and when to report earnings to maintain compliance with UI regulations.

Unemployment Insurance (UI) provides temporary income to eligible workers who lose their jobs through no fault of their own. However, *UI exhaustion* occurs once an individual has received all the benefits for which they are eligible in a given benefit year. At that point, no further payments are made unless the claim is renewed or the individual qualifies for extended benefits (Institution, 2020; on Budget & Priorities, 2021).

The actual duration of UI varies significantly by individual and state, influenced by the following factors: UI is typically calculated as a percentage of previous wages, and individuals with higher earnings get paid more. Also, most U.S. states allow up to 26 weeks of standard UI benefits. However, during economic downturns or recessions, extended benefits may become available (on Budget & Priorities, 2021).

Importantly, UI benefits generally cease when an individual finds full-time employment. For part-time employment, the rules are more nuanced. If the individual remains underemployed and continues to meet the eligibility requirements (typically related to earnings thresholds), they may still receive partial benefits. However, there is uncertainty on what can be categorized as an eligible partial employment and even if they are still eligible, UI payments may be proportionally reduced based on the income earned (Inc., 2021).

This complexity creates an environment where individuals make critical consumption and job search decisions under uncertainty. Prior work by (Lockwood, 2020; Ganong & Noel, 2019) provides empirical evidence on consumption behavior during unemployment, revealing that the majority (approximately 75%) of unemployed households exhibit a discount factor $\beta = 0.9$ which indicates more patience.

Thus, to explore this example, for our simulations, we use $\beta = 0.9$, following their setup. All other simulation parameters are similar to §5.1.

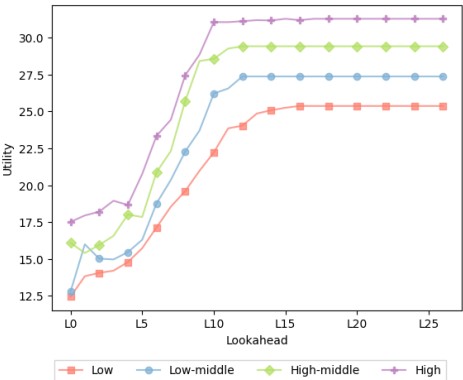

Figure 6: The final utility gained for different levels of lookahead is illustrated for the four classes, each comprising 27 unemployed individuals. Agents are of similar features, with variations solely based on the temporal aspect, i.e., the amount of lookahead in their future schedules. The $x$-axis depicts the lookahead value and the $y$-axis represents the total utility at the end of $T$ steps. Since agents are unemployed, we set $\beta$ to be 0.9 based on the prior research (Lockwood, 2020; Ganong & Noel, 2019).

**Analysis** Figure 6 illustrates the impact of foresight on utility for unemployed individuals across income levels. Three main insights emerge:

1. Individuals with little to no information about the future face significant utility losses. This is expected, as uncertainty in income, benefit eligibility, and job prospects limits their ability to plan and allocate resources effectively.

2. With increased foresight, individuals make better-informed financial decisions. The utility gain with more lookahead demonstrates that even in uncertain environments, information helps optimize consumption and savings strategies.

3. Higher-earning individuals consistently achieve greater utility, which aligns with their ability to consume without constraint. This result is evident from the upward shift in utility across the $y$-axis for the higher-earning group.

### B.3 Real-world Educational-based Patience Factors

In this section, we examine another real-world scenario: the relationship between educational attainment, consumption behavior, and the benefits of having lookahead. While the previously used discount factor of $\beta = 0.95$ in §5.1 represents a standard and commonly accepted value, our earlier analysis on unemployment illustrates that patience can vary across different demographic groups. This motivates a more context-sensitive exploration of $\beta$, particularly as it relates to educational background.

Prior empirical research demonstrates that individuals' time preferences reflected in their $\beta$ values are systematically associated with their level of education. Specifically, individuals with a high school diploma but no college degree tend to exhibit more impatient behavior, corresponding to a lower $\beta$ of approximately 0.50.

Conversely, those with a college degree tend to be more patient, with an estimated $\beta$ of 0.83 (Laibson et al., 2024; Lockwood, 2020). Therefore, in this simulation, we adopt $\beta = 0.50$ for workers with mid-level education and $\beta = 0.83$ for those with higher education, to better align our model with observed real-world behavior.

Except for this variation in $\beta$ values based on educational attainment, the experimental setup remains consistent with the baseline configuration described in §5.1. Figure 7 presents the resulting utility outcomes for workers of varying education levels.

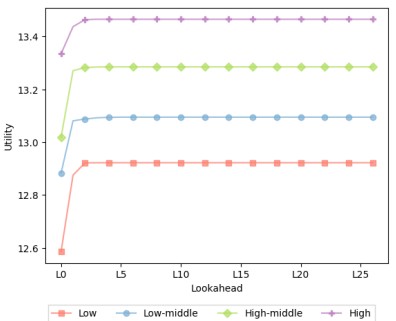

(a) High school degree lookahead analysis

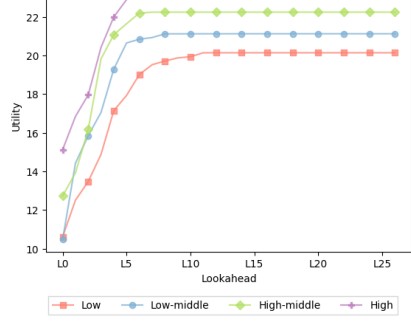

(b) College degree lookahead analysis

Figure 7: The final utility gained for different levels of lookahead is illustrated for four income classes, each comprising 27 individuals. Workers are of similar features, with variations solely based on the temporal aspect, i.e., the amount of lookahead in their work schedules. The $x$-axis depicts the lookahead value, and the $y$-axis represents the total utility at the end of $T$ steps. Here, in the left plot, the income classes have a high school (i.e., mid-level) education level with $\beta = 0.5$ and they have higher education (i.e., college degree) on the right plot with $\beta = 0.83$.

**Analysis.** The insights derived from Figure 7 can be summarized through several key observations concerning workers with lower educational attainment:

- Workers with lower education levels tend to exhibit higher levels of impatience. As a result, they reach their peak attainable utility more quickly than their more patient counterparts. For instance, individuals with only a high school education converge to their maximum utility within approximately 5 weeks, whereas college-educated workers continue to gain from lookahead information for up to 10 weeks.

- As observed in earlier settings, limited foresight leads to significantly lower utility. Workers who lack access to future information or are unable to anticipate potential instability in their schedules are less able to optimize their consumption, resulting in diminished financial well-being.

- Greater lookahead consistently improves utility outcomes. The ability to foresee and plan around future events allows workers to adjust their financial decisions accordingly (particularly within the first 5 weeks for high school-educated workers and the first 10 weeks for those with college degrees).

- Higher income levels and higher education predictably lead to higher utility even without lookahead. This is reflected in the difference in the range of the utility values along the *y*-axis in the two plots.

## B.4    Real-world Return Rates on Liquid Assets

Similar to §5.3, this section's goal is to investigate the impact of asset appreciation and depreciation on decision-making across various levels of lookahead, i.e., some workers are already at a (dis)advantage in terms of assets. While the return rates discussed in §5.3 are accurate when considering a broad spectrum of assets and realistic return ranges (Carlson, 2021; Bank, 2020), we can refine this analysis by focusing on specific categories, particularly liquid assets. Liquid assets refer to holdings that can be readily converted into cash with minimal loss in value.

In this section, we examine two of the most commonly held liquid assets: *cash* and *stocks*. The primary distinction between them lies in their nature: stocks are investment vehicles that yield returns over time, whereas cash represents direct monetary holdings. Cash does not appreciate and is vulnerable to inflationary erosion but represents the most liquid form of asset ownership.[8]

To model realistic return variability, we use empirical data from the five years preceding 2020 (i.e., 2016–2020) (Carlson, 2021; Bank, 2020). For stocks, we set a return range of 10%. For cash-only holdings, we assume the individual is holding cash only, so based on real-world returns (Carlson, 2021; Bank, 2020), we use $-0.5\%$. All other experimental conditions are held similar to 5.3.

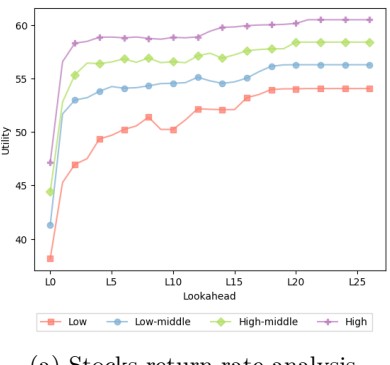

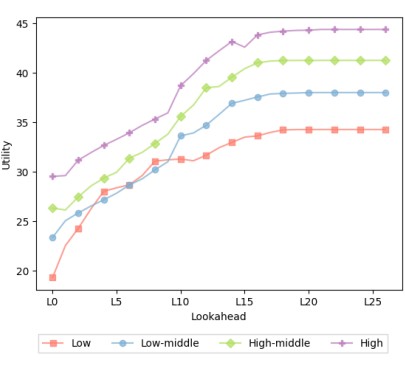

(a) Stocks return rate analysis                    (b) Cash return rate analysis

Figure 8: Individuals with similar features but varying levels of lookahead under various return rates. The return rates are set to 1.1 for stocks in the left plot. In the right plot, the return rates are set to 0.995 for cash. Other simulation parameters remain the same as 5.1.

**Analysis.**    The results, depicted in Figure 8, are consistent with those found in §5.3 and offer several noteworthy insights:

- Individuals achieve higher utility under favorable return conditions, like stocks. This trend underscores the intuitive relationship between asset performance and consumption outcomes.

---

[8]https://www.investopedia.com/articles/investing/103015/cash-vs-bonds-what-pick-times-uncertainty

- When holding a *very mildly* depreciating asset such as cash (only -0.5% depreciation), lookahead still boosts utility. Individuals quickly adapt their consumption in anticipation of diminishing purchasing power. Notably, since this asset is depreciating very mildly, the individuals attain near-maximum utility later than the original negative return setting in §5.3 and by lookahead 15.

- In scenarios with anticipated positive returns (e.g., stocks), the knowledge of a 10% increase helps individuals a lot in the beginning. However, after lookahead 5, individuals can afford to delay consumption, knowing they are likely to benefit from capital appreciation. As a result, convergence to peak utility occurs more gradually across all income groups (very late and around lookahead 20 here), reflecting increased consumption flexibility.

## B.5 More Real-world Shock Profile Case Studies

In this section, we examine another real-world scenario: the shock profiles based on the statistics provided by JPMorgan Chase Institute (2016). According to the JPMorgan Chase Institute (as explained in §5.1 as well as §A.2), income volatility remained fairly stable between 2013 and 2018, with households at the median experiencing a 36% change in income from month to month on average (JPMorgan Chase Institute, 2016). However, volatility levels vary substantially by household and occupation. The same report notes that the median standard deviation of volatility is 0.37.

Thus, we explore a case in which the shock value is selected uniformly at random from -0.36 to 0.36. We also consider the *standard deviation of volatility* (i.e., coefficient of variation as the ratio of the standard deviation to the mean), which is reported as 0.37. This is done by the coefficient of variation being defined as $\frac{\sigma}{\mu}$ where $\sigma^2$ is the variance and $\mu$ is the mean. We set the coefficient of variance to 0.37, i.e., $\frac{\sigma}{\mu} = 0.37$, which yields $\sigma = 0.37\mu$. In this case, we have $\mu = -0.36$. We consider the values in the range of $[\mu + 2\sigma, \mu - 2\sigma]$ so we end up with $-0.6264$ and $-0.0936$ as the lower and upper bounds of the range from which shocks are uniformly sampled. All other parameters are similar to §5.1.

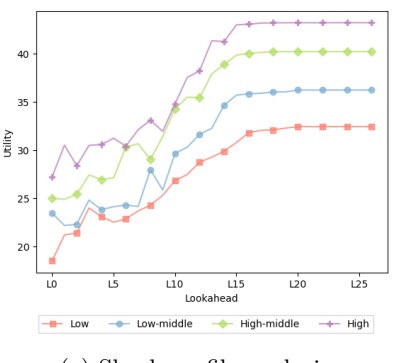
(a) Shock profile analysis

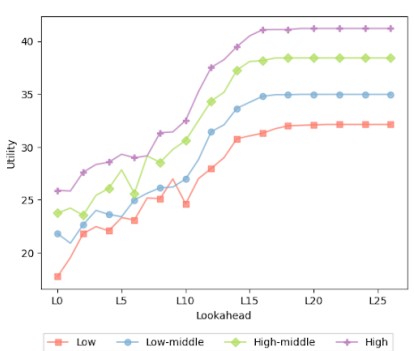
(b) Shock with standard deviation of volatility analysis

Figure 9: Individuals with similar features but varying levels of lookahead under a new shock profile. Shocks selected uniformly at random from -0.36 to 0.36 (left plot). We also consider the standard deviation of volatility, which is reported as 0.37 (right plot). Shocks are based on the exact statistics in (JPMorgan Chase Institute, 2016). Other simulation parameters remain the same as 5.1.

**Analysis.** The takeaways from Figure 9 align closely with the findings discussed in §5.2 (since the shock profiles we originally opted for were based on realistic ranges similar to the ones listed in (JPMorgan Chase Institute, 2016)):

- The same trends and consumption behaviors previously observed are present here as well, reinforcing the robustness of our results.

- Individuals with limited foresight and little knowledge about future scheduling uncertainty consistently experience lower financial utility compared to those with longer lookahead horizons.

- Greater lookahead generally leads to improved outcomes: individuals with more foresight are able to make more informed financial decisions, resulting in higher overall utility.

- Having full foresight is not strictly necessary. Once the lookahead horizon reaches around 17 weeks, utility levels closely approach those achieved under full schedule awareness.

- Increases in income correlate directly with higher utility, as they reduce financial constraints on consumption. This is clearly seen in the upward shift of the plots along the $y$-axis.

- The standard deviation of volatility introduces fluctuations, particularly affecting the two lower-income classes, which are inherently more financially unstable. However, since the reported deviation does not indicate a significant variation, the overall trend remains unchanged.

- The utility range for the deviation plot is of lower values compared to the left plot (since the shock range in the right plot was mostly negative), resulting in an overall lower utility range (as reflected in the values along the $y$-axis).

