# OpenReview forum: "Counting Hours, Counting Losses: The Toll of Unpredictable Work Schedules on Financial Security"
_TMLR — Accepted by TMLR_

### Review · Reviewer_6rip · 2025-04-27

**Summary Of Contributions:**

This paper studies the value of “lookahead”, an agent’s ability to anticipate future income shocks, in an online‐learning setting where agents face stochastic, last‐minute work‐schedule disruptions calibrated to U.S. micro-data. By varying each agent’s planning horizon, the authors demonstrate that even modest increases in foresight deliver substantial welfare gains (measured by end‐of‐horizon utility) and markedly reduce the probability of financial ruin. They then extend the framework to examine (i) how different lookahead depths interact with asset‐return volatility; and (ii) two policy interventions, pay compensation and mandatory advance‐notice mandates, showing how each can mitigate instability under realistic earnings‐shock scenarios.

**Audience:**

Yes

**Broader Impact Concerns:**

I need to put a note for the Action Editors that this particular topic falls somewhat outside my primary area of expertise; I hope my comments are taken more lightly.

In general, I think this is an interesting study with good impact for helping decision makers and policymakers invest more in helping especially working-class people build a financial risk forecasting mindset. This work seems to be of interest to the broader ML community.

**Claims And Evidence:**

Yes

**Requested Changes:**

I would suggest the authors to go through my comments in the weakness section, and I think they should address everything, apart from mapping theoretical results to simulation in the next version (i.e. **Contribution of theoretical components**).

**Strengths And Weaknesses:**

**Strengths**

* In general, well-written paper, good motivation and important problem to study
* Real-world data based simulation across income and asset distributions provides interesting insights in real-world setup. Further analysis on intervention (which comes from real world governmental policy) make these results more interesting.

**Weaknesses**

* Some of the design are not well explained in the experiments (e.g. cut of IQR, plus many other choice of numbers in the simulation). A better way is to either provide some justification from the literatures or provide more diverse simulation experiments with various setup. I feel a well documented Appendix will be necessary.

* Mapping real-world problems to experiments: I feel this paper could be strength by mapping real world problems to simulations choice (this is kind of related to the previous weakness).

* Contribution of theoretical components: I am not sure how much contribute these proofs are, they seems to be somewhat obvious - can you map some of the theoretical results to the simulation?

* Reinforcement Learning and Lookahead - this is the only technical related work section, could you expand this a bit more and maybe check some more recent papers?

---

### Review · Reviewer_WcB1 · 2025-04-27

**Summary Of Contributions:**

The submission proposes a model of utility maximization for an agent which can obtain the true values of the income and asset appreciation / depreciation over a fixed horizon. The motivation is modeling unpredictable work schedules associated with a class of jobs including gig work, food-service and retail, etc. The paper demonstrates theoretically that having access to this ground truth provides the agent with a benefit in a specific setting, and demonstrates a similar result by simulation in a more general case. Finally, it uses the model to simulate potential interventions including increasing lookahead and compensating workers for schedule changes.

**Audience:**

Yes

**Broader Impact Concerns:**

I don't have specific ethical issues with the work that I want to call out. However, considering the work's motivation as it relates to a real set of problems in current work scheduling and financial instability, I think a broader impact section would be helpful to add. In some sense, a lot of the present motivation in section 1 could be a broader impact section, and the introduction could focus more on the technical aspects of the work.

**Claims And Evidence:**

No

**Requested Changes:**

See above, but most importantly:

* The paper should clarify its engagement with other technical ML / stats contributions as relevant. At minimum I think this should include more correct engagement with H-step lookahead RL and the work by Merlis and colleagues cited above.

* The proof claims should be moderated, and more intuition given on which claims are w.l.o.g., which are technicalities, and which are core to what's doing the work.

* Missing detail in the simulation should be added (as discussed above) such that the work is properly reproducible. It's fine if some info is in supplementary material, but reading the code should be a last resort, not a requirement to understand things like distributions of various variables or discretization of the space for dynamic programming.

* Claims w.r.t. simulations should be supported by statistical hypothesis testing unless so many simulations are performed that error is negligible.

**Strengths And Weaknesses:**

**Strengths**: the submission is concerned with a real-world problem, and it motivates this problem via a variety of sources including academic literature, news sources, and government publications. This is unusual in machine learning publications and helps situate this potentially new domain.

**Weaknesses**: I have both framing and technical concerns about the work. With respect to framing, the contribution does not "read" like a machine learning paper. Most of the prior work cited comes from the economic literature, the engagement with the ML literature (notably, RL) appears incorrect to me, and the notation is imprecise. With respect to the technical contributions, I find that I have to do a lot of guessing to infer what was actually done and what to make of it, the formal results seem overstated, and I'm not sure the analysis is fully correct or supported.

I will expand on the core issues next. I did not go through for detailed minor comments and line edits as I think the paper needs a major rewrite at minimum which will likely change those minor details.

## Motivation and prior work

* Lookahead planning in ML usually implies taking an expectation over future states and returns given the current model. This is true both in e.g. H-step lookahead RL (as cited in the present contribution), lookahead in Bayesian optimization (e.g. https://arxiv.org/abs/1510.06299), and similarly in model-predictive control. By this definition, the stochastic model in section 3.1 already has "infinite" lookahead. The usual difficulty in this setting is having a good enough model for planning (in the current paper this would be a good estimate of $R_t$), plus efficiently managing the curse of dimensionality associated with taking the expectation over possible future realizations of states and actions. The submission claims in the related work section that "the idea of H-step lookahead in RL is similar to the lookahead employed in our models" but as noted above I think it's different in important ways. Furthermore, the claim "our models operate with no explicitly defined states" is also odd to me -- states are a sufficient statistic of history that decisions are conditioned on, i.e. the state is the current value of assets.

* One recent line closer to the present contribution is work by Merlis and colleagues where an agent is provided with ground truth rewards (https://arxiv.org/pdf/2406.02258, https://arxiv.org/pdf/2403.11637). Here the term lookahead is used similarly to the present contribution (and note that one of the above papers considers the case where reward but not dynamics are given, as discussed in footnote 1). In this work, the idea that access to future states is beneficial is taken is a given, and the goal is to quantify the magnitude of benefit given the additional difficulty of computing this type of lookahead policies. That work is within the broader field of competitive analysis of online algorithms, where the goal is to investigate how well an algorithm which *doesn't* have access to the true realizations can do (https://csaws.cs.technion.ac.il/~rani/book.html). Also possibly related is work on the prophet inequality, which bounds how well an agent which knows distributions of values can do relative to an agent which knows the actual inputs (https://www.sigecom.org/exchanges/volume_17/1/CORREA.pdf).

* Another related setting is in semi-autonomous agents which can query an oracle (usually a human) for some information about the evaluation environment (e.g. https://proceedings.mlr.press/v33/cohn14.pdf). Here the challenge is in determining what oracle information to obtain, and how to act if delay exists in query-answering.

I'm not sure if all of the above work needs reviewing in the present contribution, but this is simply what I found from a few minutes of chasing potential connections. TMLR does not require novelty or significance, but for an ML audience to find interest in the contribution, it should be situated relative to other technical ML contributions.

## Technical contributions

* The overall presentation could be improved. For example, the equations in 3.1. should clarify what is being optimized w.r.t. to (e.g. $\max_{c_t}$) since that is the controllable input. The notation switches from using $a_t$ for assets in section 3.1 to using $x_t$ starting in section 4. Figure 2 is given subfigures a and b but Fig 3 subfigures are unlabeled. Equations are numbered starting on page 7 but not earlier (even not earlier in Thm. 1).

* Section 4 claims that the finding holds *even* without discount and without temporal discounting and asset appreciation / depreciation, and the remark after lemma 2 states that the proof relies on no discounting, and lookahead can be made more significant with discounting. I don't think this is supported -- the proof seems to rely on the fact that without discounting and no benefit / drawback to savings, one optimal policy is to always consume all the assets (I think in this setting and with finite horizon, any policy that consumes all the assets at any point is optimal). The lookahead agent can anticipate the next income and consume it immediately, and the no-lookahead agent needs to wait one step and then consume. Over a finite horizon, this means that the no-lookahead agent will lose out on the final step's utility because it cannot anticipate consuming the final income. Next, in the setup, the expected total income changes as a function of the amount of lookahead (since income before $k$ is $Y=1$ and income after $k$ is $x\cdot Y$ where $x \sim \mathcal{U}(0,1)$). This artificially disadvantages the no-lookahead agent as lookahead increases, and seems like a weird setup at best (and at worst, is the source of the bound scaling with k). I haven't delved into the full proof in detail but this setup seems problematic. At the very least, the paper should moderate its claims and restrict them to the actual proof, provide better intuitions, and explain what's "doing the work" and what is a technicality.

* It's not clear how the algorithm works. I imagine that the setup is something like finite-horizon backward induction over a discretized [time,assets] space. Next, in order to be able to compute the probability mass assigned to each cell, I imagine one would have to compute the CDFs of the income and return distributions. Is this correct?

* The distributions of various terms are not made explicit. Section 5 describes a range and added variance of returns -- does this imply uniform + Normal additive returns? Or a truncated Normal? Section 5.1 additionally describes "shocks" with a magnitude that is a function of income and drawn from some distribution -- does this imply that income is otherwise a constant and drawn from the population for each simulation? What is the shock probability?

* Section 5.4 describes L0+Money as compensating individuals for schedule changes and on-call shifts. Is this simply adding a fixed constant to income whenever a shock occurs as described in 5.1? What is the value added? How robust is the finding against different choices of this value, specifically given finding 2 in the Analysis paragraph of section 5.4 claiming that compensation for schedule changes is not as good as lookahead? I imagine this claim would only hold for some relationship between amount of compensation and amount of lookahead (e.g. if compensation on shocks is greater than the upper bound of income, then for every step where there is a shock, L0+money will outperform lookahead on that step, and if that benefit is greater than the benefit from knowing lookahead investment returns, L0+[enough money] should outperform any lookahead without compensation). To try to understand what was actually done I tried looking through the code but even though the files are ipynb, the simulations in them are minimally-documented C code so I did not dig very deeply.

* No uncertainty is quantified for Figures 1 and 2 even though simulations are stochastic. Figure 1 claims that different income levels need different amounts of lookahead to achieve optimal utility but the difference looks numerically tiny (all lines basically saturate by L16 or so) and there are neither error bars nor hypothesis tests to support this finding. The same claim is made in section 5.3 and it is similarly unsupported. When uncertainty is quantified, it is not correct, e.g. the caption for Fig 3 describes the right side error bras as "2 standard deviations from the mean (95% CI)". It is true that for a known mean and variance, the *prediction interval* corresponds to approximately 2 standard deviations away from the mean. The CI is something about how well we know the mean (hence the CI for the mean given that the mean is known would be trivial), PI is something about a range within which future observations are likely to fall (hence if we know the mean and variance we can say something about what draws we might get). For the figures in the paper, the mean and variance are both estimated from the simulations, and we should probably be looking at the CI (which is a function of the sample variance, confidence level, and the number of observations) unless the number of simulations is large enough to shrink the CI to just be some small epsilon away from the mean.

* The benefit of lookahead seems nonmonotonic in the amount of lookahead, notably in figures 1 and 2b. This is surprising (an agent with L5 can do at least as well as L4 by ignoring the information from step 5). I am assuming at least in Fig 1 this just noise is due to insufficiently many simulations ran, but especially in Fig 2b the pattern is the same across all income classes, which looks odd and makes me suspect a bug.

---

### Review · Reviewer_qHqv · 2025-05-01

**Summary Of Contributions:**

This paper investigates the impact of unpredictable work schedules on financial stability, particularly for vulnerable worker populations. The authors introduce an online learning simulation framework to model how workers adapt consumption strategies under uncertainty and demonstrate that foresight, or having a lookahead into future schedules, significantly improves workers' financial outcomes.

According to the authors, key contributions are as follow:

- **Online Learning Algorithm with Lookahead**: The paper introduces a novel online algorithm that handles varying levels of lookahead. This algorithm helps individuals use future schedule information to make real-time consumption decisions and adapt to financial shocks. It serves as a valuable tool for studying how different degrees of foresight affect decision-making.
- **Formal Analysis of the Effects of Lookahead**: The study presents theoretical results concerning consumption behavior and the "power of lookahead". It is shown that workers who have lookahead benefit from an advantage that increases proportionally with the magnitude of their lookahead when compared to workers with no lookahead.
- **Empirical Analysis of Future Information Provision**: The paper includes an empirical exploration of how future information aids in financial management and contributes to increased utility. Experiments using the online learning algorithm investigate the effects of lookahead under uncertain job timetables.
- **Temporal Equity in Workplace Schedules**: The concept of temporal equity is explored, focusing on how the lack of advance notice (future lookahead) in work timetables acutely affects disadvantaged subpopulations.
- **Mitigation Strategies**: The paper explores various mitigation strategies, referencing fair workplace laws and acts, to understand how the negative effects of just-in-time work schedules on workers' utility can be lessened.

**Audience:**

Yes

**Claims And Evidence:**

Yes

**Requested Changes:**

Based on TMLR's acceptance criteria, I have an overall favorable opinion of this paper. However, as noted in the Weaknesses section, I also have several concerns that need to be addressed. To ensure eventual acceptance, I would like to ask your views on these concerns, including appropriate revisions to the paper.

**Strengths And Weaknesses:**

### Strengths

- **Development of a Novel Online Learning Framework with Lookahead**: A core strength is the introduction of a novel online learning algorithm designed to handle varying levels of future lookahead. The framework's ability to model different lookahead levels makes it a powerful tool for systematically investigating how varying degrees of foresight influence financial decision-making in the context of unstable work schedules. The paper also provides some theoretical results establishing the advantage conferred by lookahead.
- **Direct Relevance to Temporal Equity**: The paper effectively connects its technical framework to the pressing societal issue of temporal equity in workplace scheduling. This focus grounds the research in a critical real-world problem with significant social implications.
- **Comprehensive Review of Related Work**: The paper provides a systematic review of the relevant literature. The authors effectively contextualize their work within the existing body of research, clearly articulating its position.
- **Exploration and Evaluation of Mitigation Strategies**: The simulation framework is effectively used as a "sandbox" to explore and evaluate potential mitigation strategies for schedule instability. The analysis investigates interventions like compensation for changes and mandatory minimum advance notice periods.

### Weaknesses

**[Limited Scope of Analysis]**

- One of the key contributions of this paper is its ability to provide analytical insights into lookahead through the proposed framework. However, both the theoretical analysis in Section 4 and the experimental analysis in Section 5 appear to be limited to specific cases and lack comprehensiveness, particularly in Section 4. For instance, the latter part of Section 4 discusses the case where $t=k/2$, but a more general treatment of this scenario would be beneficial. Given the simplicity of the theorem and lemma proofs, extending the analysis should be relatively straightforward.
- Additionally, considerations beyond $\beta=1$, $R=1$, or $y=1$ should be explored more broadly if feasible. Alternatively, if such limitations are common in prior research, explicitly acknowledging this would provide useful context. Similarly, Section 5 would benefit from more extensive numerical experiments or, at the very least, a clear rationale or precedent explaining the selection of parameter settings (e.g., how the shock generation aligns with real-world conditions).

**[Validity of Assumptions]**

- Related to the above, one of the key concerns is the lack of explicit justification for the fundamental assumptions underlying the proposed mathematical model. While assumptions are an essential component in modeling complex phenomena, their validity should be clearly motivated and supported by empirical evidence, theoretical foundations, or references to established literature. In the current presentation, it remains unclear why these specific assumptions were chosen. For instance, in Section 4, I feel that $x$ being uniformly sampled and each individual knowing this distribution seems like an unrealistic assumption. Without such clarifications, readers may find it difficult to assess the robustness and relevance of the proposed framework.

**[Interpretation of Numerical Experiments]**

- The lack of explanation regarding the numerical experiments makes it difficult to address several key questions. For instance, why does the total utility (y-axis) in Figures 1 and 2 not increase monotonically as the lookahead value (x-axis) increases? What is the number of trials conducted in each experiment? Are Figures 1 and 2 based on averaged results? How statistically significant are these findings?
- Additionally, while the authors assert the robustness of experimental parameters and results, no clear evidence or simulation results are provided to substantiate this claim. Including such details would strengthen the validity of the conclusions drawn.

**[Inconsistent Symbol Definitions]**

- Throughout the paper, the same symbols appear to be defined differently in various sections, making it challenging for readers to follow the logic and mathematical formulations cohesively. For instance, the symbols for assets and consumption are denoted as $a_t$ and $c_t$ in Section 3, but are abruptly changed to $x_t$ and $z_t$ in Section 4. Even though the subscript $t$ has disappeared, $c$ and $x$ are also used with different meanings. $x\sim_{\text{uar}}(0,1)$ is probably undefined. To improve clarity and rigor, I recommend carefully reviewing all symbol definitions and ensuring uniformity throughout the paper. A consistent notation system will significantly enhance readability and prevent potential misinterpretations.

---

### Comment · Action_Editor_Ukdq · 2025-05-25
**Please engage with the author responses**

Hello reviewers,

We're have largely passed (my fault for not alerting you to this earlier) the phase of this paper's submission where you are expected to engage with the authors and address their comments as well as submit an official recommendation.

I appreciate your efforts in reviewing this paper but we are not done yet. I will not acknowledge any reviewer's recommendation who has not responded to the authors' follow-ups and done so respectfully and in good faith. We owe them this as it is a shared expectation for any of our own work. Please, at your earliest convenience, discuss with the authors whether their clarifications have addressed your concerns or if there are still outstanding issues you have with their work. The authors have also submitted a revised version of the paper that addresses several of the concerns raised in your review.

Thanks, AE

---

### Decision · Action_Editor_Ukdq · 2025-06-20

**Recommendation:** Accept as is

**Additional Comments:**

The authors were very patient with us and our efforts to fully and thoroughly evaluate their work. They were extensively responsive to reviewer comments and suggestions, providing significant improvements to their paper with each round of discussion. As such, this paper is ready, as is, for publication.

**Audience:**

Yes

**Audience Explanation:**

One reviewer articulated the challenge in reviewing this paper (from an ML frame of reference very nicely):
> Interdisciplinary / new areas often feel like square pegs in round holes when they show up in AI / ML, and this paper definitely still reads like it is coming from a different community (maybe somewhere in econ?). But I think if I view this as a new problem setting for ML, it's of potential interest and value

This paper clearly articulates the challenges it sets out to address computationally and provides well supported and principled mathematical analysis to ground the modeling approaches it uses. There is a robust community of MLxEcon researchers that should find this paper interesting and may further discussion and inquiry into ways to better account for and/or model the uncertainty, and possible interventions for, around expected income for those with unstable work schedules.

**Claims And Evidence:**

Yes

**Claims Explanation:**

The paper clearly establishes the setting in which they empirically evaluate the economic impact of individuals having better foresight of their potential earnings. This is modeling using an online planning method which is used to demonstrate the claims set forth by the authors. The reviewers were somewhat uncertain about the empirical support provided by the paper initially but the authors have greatly improved the paper to address this concern.